# OpenReview forum: "MIP against Agent: Malicious Image Patches Hijacking Multimodal OS Agents"
_NeurIPS.cc/2025/Conference — NeurIPS 2025 poster_

### Official Review · Reviewer_8zsk · 2025-06-25

**Clarity:** 3
**Significance:** 2
**Originality:** 2
**Rating:** 4
**Confidence:** 3

**Summary:**

This paper investigates attacks on vision-language model-based OS agents, which execute APIs and thus pose significant security risks. The proposed attack involves using malicious image patches (MIPs) to induce the agent to invoke specific APIs. Experimental results demonstrate that MIPs are highly effective and generalize across different user prompts and screen configurations, revealing critical security vulnerabilities in OS agents.

**Questions:**

OS agents typically require multi-step execution. Have you observed any differences in the difficulty of launching an attack depending on whether the target objective involves more or fewer steps?

**Ethical Concerns:**

["NO or VERY MINOR ethics concerns only"]

**Final Justification:**

My concerns are addressed and I keep my original rating.

**Limitations:**

Yes

**Paper Formatting Concerns:**

Format looks good

**Quality:**

2

**Strengths And Weaknesses:**

Strengths:
1. The topic of security of OS agent is very crucial, since OS agent is able to interact with real world.
2. The paper is well-written, explaining how OS agent works and how the attack constraints are derived from OS agent working pipeline.
3. The experiments are well-executed.

Weakness:
1. The robustness of the attack is not assessed: For example, when taking the screenshot, typically the image is compressed. Therefore, I suggest assessing whether the attack is still successful under common image corruptions, such as JPEG compression. Since one application of this attack requires posting the adversarial image online, experiment settings of relevant works on VLM data poisoning (which can work via posting poisoned images online) can be borrowed here, such as in [1]

[1] Xu, Yuancheng, et al. "Shadowcast: Stealthy data poisoning attacks against vision-language models." Neurips 2024.

---

> ### Author Rebuttal · Authors · 2025-07-31
>
> We sincerely thank Reviewer 8zsk for their thoughtful assessment of our work. We are especially encouraged by their perception that “*the topic of security of OS agent is very crucial*”, “*the paper is well-written*” and the “*experiments are well-executed*”.
>
> Below, we would like to address the reviewer’s main concern and the three specific questions raised.
>
> ---
> > **W**: The robustness of the attack is not assessed: For example, when taking the screenshot, typically the image is compressed. Therefore, I suggest assessing whether the attack is still successful under common image corruptions, such as JPEG compression. Since one application of this attack requires posting the adversarial image online, experiment settings of relevant works on VLM data poisoning (which can work via posting poisoned images online) can be borrowed here, such as in [1]
>
> We appreciate the reviewer’s insightful observation. We agree that robustness to image transformations is essential for assessing the practical viability of MIPs in real-world deployments, since the adversary can only place MIPs on the original screen, which is then processed by the OS agent in ways the adversary cannot control.
>
> Importantly,  **we already account for preprocessing steps applied by the OS agent before the screen is passed to the VLM**. Concretely, we define a function $q$ to capture the transformation pipeline as part of the VLM component (see lines 165–168), and we include $q$ directly in the optimisation objective (see Eq. 2). During optimisation, non-differentiable components of $q$ are then to be approximated with differentiable alternatives. Specifically, the OS agent used in the Windows Agent Arena applies a PIL-based resizing operation, which we replace with a differentiable resizing function to enable gradient-based optimization. During evaluation, however, we use the original transformation pipeline, faithfully simulating a realistic deployment of the OS agent. Our results show that MIPs crafted in this way remain robust when placed on the original screen, even after being processed by the OS agent’s transformation pipeline.
>
> We acknowledge that if the OS agent’s pipeline were later modified to include further transformations not modeled in $q$, this could affect the robustness of MIPs. That said, adding such preprocessing steps may also degrade the performance of the screen parser or the VLM itself, introducing a trade-off between robustness and performance. We believe further exploring this is a valuable direction for future research on MIP defenses. We will update our discussion section to include common image corruptions as a potential mitigation strategy.
>
> ---
> > **Q**: OS agents typically require multi-step execution. Have you observed any differences in the difficulty of launching an attack depending on whether the target objective involves more or fewer steps?
>
> In the paragraph “*Transferability of MIPs Across Execution Steps*” in Section 4.2, we evaluate the performance of MIPs when they are encountered by the OS agent during the execution of a benign multi-step task. As stated in line 323, the OS agent encounters the MIP after 1 to 10 steps, with at least two out of three or one out of three attacks being successful when placed on a seen or unseen screen configuration, respectively. However, we did not observe a clear correlation between the number of steps taken and the attack success rate. This suggests that **MIP effectiveness is largely insensitive to the specific execution step** at which it is encountered.
>
> ---
> Finally, we would like to thank the reviewer again for their valuable feedback which helped to further improve the manuscript. We hope our answers could clarify all remaining questions and reaffirm the merits of our work.

---

### Official Review · Reviewer_o8pU · 2025-06-26

**Clarity:** 3
**Significance:** 3
**Originality:** 2
**Rating:** 4
**Confidence:** 5

**Summary:**

This work explores the extension of adversarial image patches to the OS agent setting. The authors demonstrate via empirical experiments that it is possible to craft malicious image patches that induce the agent to perform malicious actions. The methodology employed by the authors extends existing projective gradient approaches, and a variety of ablations are applied to demonstrate the robustness of the approach.

**Questions:**

- Fig 2: please elaborate on what parts of the images are MIPs
- The background may not always be visible - can other areas be targeted such as application icons in the task bar?
- It might be unrealistic to expect access to the system prompt and some evaluations of ASR over variations in the prompt or perhaps removal of the system prompt entirely would be valuable. My intuition is that the ASR should remain high since the MIP may act to suppress any attention to the input text tokens.

- The work is very similar in flavor to https://arxiv.org/html/2407.08970v1 which I don’t see in the citations. Like this work, that paper uses PGD to optimize input images to inject meta-instructions into the output, overriding the user request and yielding potentially damaging output. This work mainly extends that approach to an agentic setting where a) the attack may be triggered in a multi-step workflow, b) the stakes are considerably higher, and c) some additional factors, such as image parsing, may impact attack success. Please comment on whether this is an accurate characterization.

**Ethical Concerns:**

["NO or VERY MINOR ethics concerns only"]

**Final Justification:**

Thank you for addressing my comments and questions.
I want to clarify that my thoughts on invariance to changes in the system prompt were largely speculative and it would be best to back up these assertions with some empirical results. It is helpful to know that you used the standard system prompt from WindowsAgentArena, which we might expect some number of deployments to use unchanged.

Having read your responses as well as the other reviews and rebuttals, especially the general consensus among reviewers around limitations, I'm electing to keep my rating at borderline accept.

**Limitations:**

yes

**Quality:**

3

**Strengths And Weaknesses:**

Strengths:
- The problem is of critical interest to the community, as there are many open questions about the safety of deploying computer-using agents.
- The experimentation demonstrates high attack success and reasonably strong invariance across previously unseen screenshots in the universal setup.
- The approach of choosing a sub-region of the image is successful and makes the attack more broadly applicable to settings where the adversary may only have limited control over the screen pixels.

Weaknesses:
- The approach requires white box access to the target model, and also requires knowledge of the system prompt.
- Only two malicious targets were explored. How were these selected?
- The threats posed by the work merit further discussion of potential mitigations

---

> ### Author Rebuttal · Authors · 2025-07-31
>
> We sincerely thank Reviewer o8pU for their thoughtful assessment of our work. We are particularly encouraged by the reviewer’s recognition that “*problem is of critical interest to the community*”, that the“*experimentation demonstrates high attack success and reasonably strong invariance*”, and that our “*attack has broad applications*”.
>
> Below, we address the reviewer’s main concerns and the four specific questions raised.
>
> ---
> > **W1**: The approach requires white box access to the target model […]
>
> We appreciate the reviewer’s insightful observation. To clarify, while it is true that the optimisation procedure assumes white-box access to an underlying VLM component of the OS agent, the evaluations are conducted under “black-box conditions” as well. Specifically, we assess the generalisability of MIPs to realistic deployment scenarios along several axes of variation, including: unseen user prompts, unseen screen layouts, unseen screen parsers, unseen preceding execution steps, and different temperature settings**.
>
> We explicitly designed the experimental setup to explore where MIPs generalise and where they do not, to provide a comprehensive understanding of their capabilities and limitations. We find that MIPs successfully generalise across all combinations of these unseen variations mentioned above. Furthermore, while MIPs also transfer to multiple VLMs used during optimisation, we observe that they do not generalise to entirely unseen VLMs. We openly discuss this limitation and its implications in the paper (see lines 343–351). However, we note that many real-world applications are currently based on open-source models, making gray-box access a concrete and actionable threat.
>
> ---
> > **W1**: [...] and also requires knowledge of the system prompt.
>
> > **Q3**: It might be unrealistic to expect access to the system prompt and some evaluations of ASR over variations in the prompt or perhaps removal of the system prompt entirely would be valuable. My intuition is that the ASR should remain high since the MIP may act to suppress any attention to the input text tokens.
>
> We acknowledge that in our experiments, we used the default system prompt $p_{\text{sys}}$ from the Windows Agent Arena and kept it fixed throughout. However, we would like to emphasise that we thoroughly evaluated the generalisability of MIPs across variations in the user prompt $p$, the screen parser description $p_{\text{som}}$, and the agent memory $p_{\text{mem}}$, all of which contribute substantially to the textual input to the VLM.
>
> Given that MIPs maintain a high ASR despite significant variations in the textual input, we agree with the reviewer’s intuition that changing or even removing $p_{\text{sys}}$ would likely have minimal effect on ASR. This supports the idea that MIPs effectively “*suppress any attention to the input text tokens*”. We added this clarification to the final version of the paper to better reflect this insight.
>
> ---
> > **W2**: Only two malicious targets were explored. How were these selected?
>
> We deliberately chose two realistic and general malicious behaviours that reflect plausible, high-impact threats within the OS agent framework. These behaviours are representative of the types of actions that, if triggered by a MIP, could result in real-world harm.
> In selecting the two malicious targets, we also aimed to evaluate how output sequence length affects MIP effectiveness, with the first being a shorter 33-token-long output and the second being a longer 52-token-long output. This design allowed us to examine whether sequence length influences attack success rates or robustness.
> As noted in lines 279–281 of the paper, we assume that if MIPs can reliably trigger both of these diverse malicious behaviours, it provides strong evidence of their ability to generalise to a broader class of malicious actions executable by the OS agent. We view this as a principled starting point for demonstrating the feasibility and generality of MIP-based attacks.
>
> ---
> > **W3**: The threats posed by the work merit further discussion of potential mitigations
>
> We appreciate the reviewer’s attention to potential defenses. While our primary objective in this work is to systematically expose a novel and practical vulnerability in multimodal OS agents, we agree that mitigation is an important direction. We are actively exploring such defenses in follow-up work, which we see as orthogonal and complementary to the core contributions of this paper. Nonetheless, in line with the reviewer’s suggestion, we extend the discussion of potential defense strategies in Section 5 of the final version.
>
> ---
> > **Q1**: Fig 2: please elaborate on what parts of the images are MIPs
>
> We kindly refer to Figures 6 and 7 in the appendix, where we explicitly illustrate the MIPs that were embedded in the two respective settings used to trigger the malicious behaviours.
>
> ---
> > **Q2**: The background may not always be visible - can other areas be targeted such as application icons in the task bar?
>
> Theoretically, it is indeed possible to craft MIPs for any image patch region, including application icons in the task bar, as long as the adversary has access to change their visual content.
>
> To highlight the practicality and reach of the threat, we deliberately focused on more publicly accessible and general image patch regions, such as uploaded content on social media, that an attacker can realistically influence and embed MIPs into without requiring privileged access. These MIPs can then propagate online and persist in public content, waiting to be captured by OS agents during regular use.
>
> ---
> > **Q4**: The work is very similar in flavor to https://arxiv.org/html/2407.08970v1 which I don’t see in the citations. Like this work, that paper uses PGD to optimize input images to inject meta-instructions into the output, overriding the user request and yielding potentially damaging output. This work mainly extends that approach to an agentic setting where a) the attack may be triggered in a multi-step workflow, b) the stakes are considerably higher, and c) some additional factors, such as image parsing, may impact attack success. Please comment on whether this is an accurate characterization.
>
> We thank the reviewer for pointing out this related work. The reviewer is correct that our work extends the threat to a system-level, multimodal OS agent setting, introducing additional pipeline components, stricter constraints, multi-step execution, and higher real-world stakes.
>
> Crucially, our attack is also more direct: the entire malicious behaviour is explicitly encoded in the adversarial patch, meaning that once the OS agent is hijacked, the attack executes immediately without relying on the agent to correctly interpret the meta-instruction and generate malicious actions on its own. This reduces failure modes and increases the reliability of MIPs in practice.
>
> We add this comparison to the related work section in the final version of our paper.
>
> ---
> We once again thank the reviewer for their thoughtful feedback, which has contributed to further strengthening our work. We hope that our clarifications address all remaining concerns and reinforce the value of our contribution.

---

> ### Comment · Reviewer_o8pU · 2025-08-05
> **response to authors**
>
> I'm pasting my final justification here since at the time I submitted I didn't realize it wasn't visible to authors.
>
> --
>
> Thank you for addressing my comments and questions. I want to clarify that my thoughts on invariance to changes in the system prompt were largely speculative and it would be best to back up these assertions with some empirical results. It is helpful to know that you used the standard system prompt from WindowsAgentArena, which we might expect some number of deployments to use unchanged.
>
> Having read your responses as well as the other reviews and rebuttals, especially the general consensus among reviewers around limitations, I'm electing to keep my rating at borderline accept.

---

> > ### Author Response · Authors · 2025-08-06
> >
> > Thank you for the additional clarification.
> >
> > Although the limited time remaining in the rebuttal period did not allow us to finalize the ablation study on system prompt invariance, initial results support our intuition: variations in the system prompt (alongside the already evaluated variations in textual input such as the user prompt, screen parser description, and agent memory) seem to have minimal effect on the ASR of the MIPs. We will include this additional ablation study in the final version of the paper to substantiate these findings.
> >
> > Overall, we truly appreciate the positive assessment of our work.

---

### Official Review · Reviewer_LVfM · 2025-07-03

**Clarity:** 4
**Significance:** 4
**Originality:** 4
**Rating:** 6
**Confidence:** 4

**Summary:**

This paper proposal a novel attack method targeting multimodal OS agents -- "Malicious Image Patches". Since OS agents rely on screenshots for environmental perception and decision-making, attacker can create an image patch containing adversarial disturbances imperceptible to human eye. When this patch is captured by the OS agent, it can directly hijack the agent's underlying VLM, causing it to ignore normal user commands and instead generate and execute malicious code preset by the attacker. The authors formalize the generation of MIP as an optimization problem with multi-constraints. The optimization process aims to find a perturbation $\delta$ that maximizes the probability of the VLM generating specific malicious instructions, without changing the screen parser's output, covering interface elements, and under the constraint of limited perturbation magnitude.

This paper demonstrates that MIP attacks are not only effective in specific scenarios but, more importantly, specially optimized "universal MIPs" can generalize to unknown user instructions, different screen layouts, and different screen parser components, even successfully triggering in the middle of multi-step task execution by the agent. Additionally, the paper shows that through joint optimization, a single MIP can simultaneously attack multiple different-sized and types of VLMs within the same model family (such as Llama-3.2 11b and 90b), greatly enhancing the real-world threat of the attack.

**Questions:**

# Regarding the robustness of the invariance assumption for the screen parser $g(s) = g(s + \delta)$

The key premise of the entire optimization process is that the added adversarial perturbations $\delta$ do not change the output of the screen parser $g$ . You mentioned that this assumption usually holds in practice, but you did not provide any quantitative evidence. This raises a question: where is the boundary for the validity of this assumption? For example, at what $\epsilon$ value would it fail? Or, does this assumption still hold on complex screens containing a large amount of dense text or UI elements? If $g(s) ≠ g(s + \delta)$ , would the attack fail immediately? Because at that point, the p_som and s_som received by VLM would change, thus destroying the carefully optimized input.

Suggestions: It is recommended that the authors conduct an ablation study: after successfully generating MIP, deliberately fine-tune $\delta$ until $g(s) ≠ g(s + \delta)$ , then test the change in attack success rate. Or, statistically analyze how many percentage of gradient update steps in the optimization process would violate this constraint. If the authors can prove that the method is not sensitive to this, or that the attack success rate decreases smoothly rather than dropping to zero abruptly when the constraint is broken, this would greatly enhance the reliability of the methodology. Conversely, if the attack is extremely sensitive to this, then it is an important limitation that needs to be acknowledged.

# About the patch placement constraints $s_{\text{som}} \odot \mathbb{1}_{\mathcal{R}} = 0$


Your constraint requires that the malicious patch R cannot overlap with any UI elements detected by screen parsers. This is a very strong simplification. On real, highly dynamic, or cluttered screens (e.g., a desktop with multiple open windows and notifications), it may be difficult to find a sufficiently large and "clean" area R to embed the patch. The places where attackers most want to embed the patch (such as ad banners, the top of social media image feeds) are exactly the ones that may overlap with UI elements. Can your method be extended to handle this situation?

Suggestions: Could this constraint be relaxed, for example, by allowing overlap with certain "non-critical" UI elements (such as plain text labels) and adding a regularization term in the optimization objective to minimize the semantic impact on these elements? Exploring this issue will directly relate to the generality of this attack in the real world.

# Related to threats to black-box models and comparisons with contemporary work

Experiments (Section 4.3) honestly demonstrate that MIPs are difficult to transfer to completely unknown VLMs. This suggests that the attack currently primarily targets white-box or gray-box scenarios (the attacker understands the architecture of the target VLM, at least the same series of models). However, many commercial OS agents may use closed-source proprietary models in the future. Does this mean the realistic threat of MIPs is mainly limited to the open-source ecosystem? Additionally, considering the work of Wu et al. [54], can you more specifically elaborate on the fundamental advantages of your "direct hijacking" method compared to their "misleading caption model" method in terms of attack capabilities (e.g., being able to generate more complex, more arbitrary malicious code)?

Suggestions: It is suggested to delve deeper into the possibility of black-box attacks (e.g., using query-based attacks or model stealing techniques to approximate the target VLM) and the limitations of current methods in the discussion section.

**Ethical Concerns:**

["NO or VERY MINOR ethics concerns only"]

**Final Justification:**

This paper presents complete experiments and comprehensive context. The authors conducted the rebuttal very seriously and responsibly, clearly addressing the concerns I raised as well as clarifying the ambiguities in the original text. Therefore, I believe this is a very thorough and complete paper, and it should be accepted.

**Limitations:**

The author has already discussed some limitations of this method in Sections 4.3 and 5, particularly its insufficient transferability to unknown VLMs. This is commendable.

There are still several limitations that have not been fully acknowledged:

1. The author mentions the direct harm of attacks but could further emphasize their potential "worm-like" effect. If the malicious act itself is "sharing this post on social media," MIP can achieve self-propagation, causing widespread impact. Explicitly pointing this out would make the discussion on negative social impact more profound.

2. The defense strategies proposed at the end of the paper (validators, consistency checks) are rather macroscopic. A more specific limitation is that current adversarial perturbations can be fragile against some simple preprocessing (such as JPEG compression, Gaussian blur). The authors did not discuss the robustness of their attacks against such common image transformations.

**Paper Formatting Concerns:**

Good paper formatting with clear tables and illustrations.

**Quality:**

4

**Strengths And Weaknesses:**

Quality

Strength: 1. Rigorous Methodology. The design of the objective and constraints is very ingenious, which reflects the author's deep understanding of the non-differential, non-continuous aspect of the OS Agent's workflow. 2. Experiment design is comprehensive and reliable. The paper systematically evaluates the generalization capability of attacks across multiple dimensions: instructions, layout, parser, execution steps, and VLM models. Specifically, the experiments in "Execution Step Transferability" (Table 3) simulate scenarios where a proxy accidentally captures MIPs during the execution of benign tasks, significantly enhancing the real-world credibility of the attack. In addition, Various temperature settings were used in the experiment for sampling evaluation, and mean and standard deviation were reported, which enhances the reliability of the results.

Weaknesses: The key assumption in the entire optimization model is $g(s) = g(s + \delta)$, that small perturb $\delta$ do not change the output of the non-differential screen parser $g$. The authors state in the text, "fortunately, this constraint is usually satisfied in practice, since ε is small by design." This is a very strong assertion, but it lacks experimental data support. If the perturbation happens to cross a certain threshold in OCR or object detection, causing the parser output ( $p_{som}$ or $s_{som}$ ) to change, the premise of the entire attack no longer holds. While this is not fatal, the lack of discussion on the boundary of this assumption constitutes a minor logical flaw. This is not grounds for rejection, but it needs to be supplemented in the final version.

---

Clarity

Strengths: clear description and clearly defined terminologies.

Weaknesses: None

---

Significance

Strength: 1. The MIP attack vectors revealed in this paper are more covert, harder to defend against, and more contagious (e.g., through social media) compared to known text prompt injection or deceptive pop-up attacks. 2. This work is almost certain to inspire a series of follow-up research, including stronger attack variants, targeted defense mechanisms (such as visual-textual consistency detection, adversarial purification, etc.), and more secure OS Agent architecture designs.

Weaknesses: None

---

Originality

Strengths: 1. The authors creatively addressed how to bypass non-differential components and how to generate payloads under multi-constraint conditions (patch location, API format).

Weaknesses: None

---

> ### Author Rebuttal · Authors · 2025-07-30
>
> We sincerely thank Reviewer LVfM for their thoughtful and high-quality assessment of our work. Their elaborate comments *reflect a deep understanding of our paper and the broader topic*, and we are especially encouraged by their positive feedback.
>
> We are pleased to address the points raised in detail below:
>
> ---
> > **Screen Parser Constraint**: The key premise of the entire optimization process is that the added adversarial perturbations $\delta$ do not change the output of the screen parser $g$. You mentioned that this assumption usually holds in practice, but you did not provide any quantitative evidence. This raises a question: where is the boundary for the validity of this assumption? [...]
>
> We appreciate the reviewer recognising this subtle but important detail, and the opportunity to further clarify the questions around the stability of the screen parser $g$ both in theory and in practice.
>
> While our formal objective in Eq. (2) enforces the constraint $(s_{\text{som}}$, $p_{\text{som}}) = g(s) = g(s + \delta)$ to rigorously preserve the screen parser’s output, in practice, our primary concern is that the screen parser does not place any bounding boxes within the image patch region $R$.
>
> This is because we can confirm that changes to the parser’s output outside of $R$ do not impact the effectiveness of MIPs, even when $s_{\text{som}}$ and $p_{\text{som}}$ vary significantly. This is supported by our generalisability results to unseen screen layouts, where different screen configurations lead to substantial variations in $s_{\text{som}}$ and $p_{\text{som}}$. This holds across both the desktop configurations, where only 18 elements are detected, and the “denser” social media configurations, where 62 elements are detected, as briefly noted in line 287.
>
> In contrast, we have observed that when parts of the MIP are covered, the attack success rate drops noticeably. Thus, to guarantee that no bounding boxes directly interfere with the MIP, our optimisation procedure explicitly checks whether the screen parser places any bounding boxes within the image patch region $R$ after every fixed number of optimisation steps. If this is violated, we roll back to the most recent valid checkpoint, apply small random perturbations, and clip the result back into the allowed $\ell_\infty$-ball, which nudges the optimisation in a new direction to prevent recurring constraint violations. In our final experiments, however, this mechanism was never triggered, indicating that for small $\epsilon$ values, the screen parser remains stable throughout.
>
> To summarise, the attack is *not sensitive to changes in the screen parser output outside the image patch region $R$, and we enforce that the parser does not interfere within $R$ by design*. We will include these implementation details in the final version of the paper and thank the reviewer for prompting a deeper discussion of this important aspect.
>
> ---
> > **Image Patch Region Constraint**: Your constraint requires that the malicious patch R cannot overlap with any UI elements detected by screen parsers. This is a very strong simplification. On real, highly dynamic, or cluttered screens (e.g., a desktop with multiple open windows and notifications), it may be difficult to find a sufficiently large and "clean" area R to embed the patch. The places where attackers most want to embed the patch (such as ad banners, the top of social media image feeds) are exactly the ones that may overlap with UI elements. Can your method be extended to handle this situation?
>
> We thank the reviewer for this insightful observation. We would like to address this point by distinguishing between the optimisation phase (where the MIP is crafted) and the runtime phase (where the MIP is captured by the OS agent).
>
> During *optimisation*, the restriction that $R$ does not overlap with dynamic UI elements is a practical constraint, since the adversary typically has control only over static images. Optimising the MIP on top of dynamic UI elements would hard-code them into the patch, which would be undesirable. Thus, ensuring that $R$ is and remains “clean” during optimisation is both crucial and realistic (as discussed above).
>
> At *runtime*, however, we acknowledge that the MIP may be temporarily occluded by dynamic UI elements. In such cases, the MIP remains embedded in the static image and becomes effective once it is revealed (e.g., in the desktop setting when windows are closed). While our current method does not explicitly account for such occlusions, we agree that enhancing robustness to partial visibility is valuable in practice.
>
> As the reviewer suggests, a promising extension would be to incorporate regularisation into the optimisation to make the MIP robust to partial occlusions. For instance, this could be achieved by adding pixel-level dropout masks at each optimisation step. While this is not yet implemented in our current pipeline, it may be an effective strategy to further improve real-world applicability. In response to this suggestion, we are conducting an ablation study on the success rate of such “occlusion-robust” MIPs, and we will include these results and a related discussion in the final version of the paper.
>
> ---
> > **MIP Transferability**: Experiments (Section 4.3) honestly demonstrate that MIPs are difficult to transfer to completely unknown VLMs. [...] Does this mean the realistic threat of MIPs is mainly limited to the open-source ecosystem?
>
> We appreciate the reviewer’s thoughtful question and agree that the limited generalisibility of MIPs to entirely unseen VLMs is an important observation. That said, we emphasise that the VLM is only one component of the full OS agent pipeline. Our experiments show that MIPs do generalise reliably across other unseen components. Furthermore, many real-world applications are currently based on open-source models, making white-box or gray-box access a concrete and actionable threat. However, we agree with the reviewer that extending our method to fully black-box scenarios is a valuable direction for future work, though it goes beyond the scope of this paper given the already complex OS agent pipeline and the associated computational demands. We follow the reviewers suggestions and delve deeper into a discussion of black-box attacks and the limitations of current methods in the discussion section.
>
> ---
> > **Hijacking Methods**: Additionally, considering the work of Wu et al. [54], can you more specifically elaborate on the fundamental advantages of your "direct hijacking" method compared to their "misleading caption model" method in terms of attack capabilities (e.g., being able to generate more complex, more arbitrary malicious code)?
>
> Regarding the comparison to “*indirect hijacking*” methods such as Wu et al. [54], our “*direct hijacking*” approach offers a key advantage: the entire malicious behavior is explicitly encoded in the adversarial patch, which means that once the OS agent is hijacked, the desired behavior is executed immediately. In practice, we have observed that “*indirect hijacks*” often introduce additional points of failure, where the agent may be compromised but the attack fails because it cannot parse or execute the intended behavior. Our approach avoids this issue by reliably triggering relatively complex malicious behavior. We included these comparisons of “direct” and “indirect” hijacks in the final version of our paper.
>
> ---
> > **Worm-like effect**: The author mentions the direct harm of attacks but could further emphasize their potential "worm-like" effect. [...] Explicitly pointing this out would make the discussion on negative social impact more profound.
>
> We thank the reviewer for pointing us towards this. In the final version of our paper, we explicitly emphasise the "worm-like" attack, where MIPs can achieve self-propagation to cause widespread impact, further discussing the broader societal risks posed by MIPs.
>
> ---
> > **Defense strategies**: The defense strategies proposed at the end of the paper (validators, consistency checks) are rather macroscopic. A more specific limitation is that current adversarial perturbations can be fragile against some simple preprocessing (such as JPEG compression, Gaussian blur). The authors did not discuss the robustness of their attacks against such common image transformations.
>
> We agree that robustness to image transformations is essential for assessing the practical viability of MIPs in real-world deployments, since the adversary can only place MIPs on the original screen. Importantly, we account for this by defining a function $q$ to capture the image transformation as part of the VLM component (see lines 165–168), and including $q$ directly in Eq. (2). During optimisation, $q$ is approximated with a differentiable alternative. During evaluation, however, the original $q$ is used to faithfully simulate a realistic deployment of the OS agent. Our results show that MIPs crafted in this way remain effective when placed on the original screen.
>
> However, we acknowledge that MIP robustness could be degraded if the transformation pipeline were later modified, such as adding JPEG compression or Gaussian blur, as mentioned by the reviewer. But this may also degrade the performance of the screen parser or the VLM itself, introducing a trade-off between robustness and performance. We believe exploring this trade-off is a valuable direction for future research on MIP defenses. We revise the discussion on limitations, especially the robustness of MIPs against image transformations.
>
> ---
> In conclusion, we thank the reviewer once again for their detailed and insightful comments. The points raised (ranging from theoretical constraints and their implications to suggestions for extensions, comparisons to related work, and broader impact) help clarify important aspects in the final version of our paper. We greatly appreciate the reviewer’s contribution to further strengthening this work.

---

### Official Review · Reviewer_QnEz · 2025-07-11

**Clarity:** 2
**Significance:** 2
**Originality:** 2
**Rating:** 2
**Confidence:** 4

**Summary:**

This paper investigates a new class of adversarial attacks—Malicious Image Patches (MIPs)—targeting multimodal OS agents, which are agents capable of performing actions on a user's computer based on screenshots and language instructions. The attack involves embedding small adversarial image patches in benign-looking images (e.g., wallpapers or social media posts). When captured in a screenshot by an OS agent, these patches induce the agent to perform harmful actions such as navigating to malicious websites or causing memory overflows. The paper presents targeted and universal attacks, demonstrates transferability across screen parsers and execution steps, and evaluates their effectiveness using the Windows Agent Arena framework and various LLaMA-based vision-language models.

**Questions:**

see weakness

**Ethical Concerns:**

["NO or VERY MINOR ethics concerns only"]

**Final Justification:**

The authors' response does not adequately address the concerns.  I maintain my score.

**Limitations:**

yes

**Quality:**

2

**Strengths And Weaknesses:**

Strengths:
1. The problem is important and timely. As OS agents become more prevalent, investigating their security vulnerabilities is highly important.
2. The adversarial objective is well-defined under realistic constraints (e.g., small patch size, non-interference with screen parser bounding boxes).

Weaknesses:
1. Limited novelty: The concept of using image patches for adversarial attacks is not new and has been widely explored in prior literature. While the paper adapts this idea to the OS agent setting, the core technical methodology (e.g., PGD under perceptual constraints) follows existing adversarial attack frameworks.
2. Assumption of white-box access: The attack relies on access to model gradients and internal agent configurations, which may limit its practicality in black-box or deployed systems.
3. Defense strategies underdeveloped: The discussion of defenses is relatively high-level and lacks concrete implementation or evaluation.
4. Lack the comparison with related work.

---

> ### Author Rebuttal · Authors · 2025-07-30
>
> We thank Reviewer QnEz for their assessment of our work. We are encouraged by their recognition that the "*problem is important and timely*" and that the "*adversarial objective is well-defined under realistic constraints*".
>
> Below, we respond to the reviewer’s main concerns in detail:
>
> ---
> > **W1**: Limited novelty: The concept of using image patches for adversarial attacks is not new and has been widely explored in prior literature. While the paper adapts this idea to the OS agent setting, the core technical methodology (e.g., PGD under perceptual constraints) follows existing adversarial attack frameworks.
>
> Our work is far and different the existing literature on adversarial attacks on image patches in prior literature by being **fundamentally different in scope, setting, and implications**.  We are not attacking an image (with minor constraints on perceptibility), we are attacking a full agentic system that all together should be fooled under many constraints. Our **system-level attack on multimodal OS agents** in the vision domain introduces unique challenges and constraints, such as:
>
> 1. MIPs must conform to the functionality of non-differentiable, non-continuous system components of the OS agent.
> 2. Furthermore, it must be ensured that they only affect targeted components while keeping the output of other components unchanged.
> 3. MIPs can only be placed in a small, predetermined region of the screen, such as an image uploaded to a social media website. They can no manipulate the entirety of the screen pixels. Additionally, they must remain effective after preprocessing of the screenshot by the OS agent, prior to being passed to the VLM.
> 4. MIPs must generalise across multiple realistic scenarios, including diverse user behaviors, different execution steps, varying screen layouts, and unknown OS agent components.
>
> These constraints increase the technical difficulty and also enhance the practical relevance of the attack vector. We are the first to systematically explore these challenges, shifting the focus from model-centric adversarial attacks to realistic, system-level attacks posed by MIPs in the vision domain. Our findings show that **MIPs introduce actionable, real-world threats with tangible consequences that are difficult to detect and mitigate**. We believe this constitutes a novel and timely contribution.
>
> While adversarial image attacks have a rich history, we respectfully but strongly disagree with the implication that applying existing methodologies (in our case just being the use of a PGD to solve part of our proposed optimization problem) to a substantially different setting diminishes the contribution’s novelty. **Adapting established techniques to new settings has repeatedly led to impactful advances in the literature* (published in tier-one conferences)* (see Kurakin et al., 2017; Zou et al., 2023; Bailey et al., 2023; Gu et al., 2024; Schaeffer et al., 2024, among others).
>
> ---
> > **W2**: Assumption of white-box access: The attack relies on access to model gradients and internal agent configurations, which may limit its practicality in black-box or deployed systems.
>
> We thank the reviewer for raising this important point. To clarify, while it is true that the optimisation procedure assumes white-box access to an underlying VLM component of the OS agent, the evaluations are conducted under “black-box conditions” as well. Specifically, we assess the generalisability of MIPs to realistic deployment scenarios along several axes of variation, including: **unseen user prompts (Tables 1, 2, 3 ,4), unseen screen layouts (Tables 1, 2, 3 ,4), unseen screen parsers (Appendix Table 8), unseen preceding execution steps (Table 3), different temperature settings (Tables 1, 2, 3 ,4), and VLM Models (Appendix Table 9)**.
>
> We explicitly designed the experimental setup to explore where MIPs generalise and where they do not, to provide a comprehensive understanding of their capabilities and limitations. We find that **MIPs successfully generalise across all combinations of these unseen variations** mentioned above. Furthermore, while MIPs also transfer to multiple VLMs used during optimisation, we observe that they do not generalise to entirely unseen VLMs. We openly discuss this limitation and its implications in the paper (see lines 343–351). However, we note that many real-world applications are currently based on open-source models, making gray-box access a concrete and actionable threat. We agree with the reviewer that extending our method to fully black-box scenarios is a valuable direction for future work, though it goes beyond the scope of this paper given the already complex OS agent pipeline and the associated computational demands. We further extend the discussion of black-box attacks and the limitations of current methods in the final version of our paper.
>
> ---
> > **W3**: Defense strategies underdeveloped: The discussion of defenses is relatively high-level and lacks concrete implementation or evaluation.
>
> We appreciate the reviewer’s interest in potential defenses. Our **primary objective in this work is to systematically expose a novel and practical vulnerability in multimodal OS agents**, not to propose or benchmark defense mechanisms. As with prior foundational work on adversarial attacks (e.g., Kurakin et al., 2017), we believe that clearly **identifying and characterising the threat is a critical first step** before rigorous defenses can be developed.
>
> That said, we currently explore such defenses as a valuable follow-up, which we view as orthogonal and complementary to the core contributions of this paper. In the final version of this paper, we are happy to extend the discussion on potential defenses in Section 5, as suggested by the reviewer.
>
> ---
> > **W4**: Lack the comparison with related work.
>
> We acknowledge the reviewer’s suggestion and would be grateful for clarification on which specific related works or comparisons are considered missing. We made a concerted effort to conduct a thorough literature review, covering both traditional adversarial image attacks and recent work on OS agents. In particular, we provide explicit comparisons to the most closely related works and highlight the conceptual and technical differences in Section 2.
>
> If there are **specific works the reviewer believes should be discussed**, we would be happy to incorporate them into the final version of the paper.
>
> ---
> In conclusion, we thank the reviewer again for their valuable feedback. We hope our responses clarify the scope of our contribution, and we believe that the additional insights gained from this exchange further strengthen the paper. Should any further questions arise, we would be happy to address them. Otherwise, we hope the reviewer will consider a positive reassessment of our work.
>
> ---
> A. Kurakin, I. Goodfellow, and S. Bengio. *Adversarial examples in the physical world*, 2017.
>
> A. Zou, Z. Wang, N. Carlini, M. Nasr, J. Zico Kolter, and M. Fredrikson. *Universal and Transferable Adversarial Attacks on Aligned Language Models*, 2023.
>
> L. Bailey, E. Ong, S. Russell, and S. Emmons. *Image Hijacks: Adversarial Images can Control Generative Models at Runtime*, 2023.
>
> X. Gu, X. Zheng, T. Pang, C. Du, Q. Liu, Y. Wang, J. Jiang, and M. Lin. *Agent Smith: A Single Image Can Jailbreak One Million Multimodal LLM Agents Exponentially Fast*, 2024.
>
> R. Schaeffer, D. Valentine, L. Bailey, J. Chua, C. Eyzaguirre, Z. Durante, J. Benton, B. Miranda, et al. *Failures to Find Transferable Image Jailbreaks Between Vision-Language Models*, 2025.

---

> > ### Comment · Reviewer_QnEz · 2025-08-06
> > **I maintain my score**
> >
> > The authors' response does not adequately address the concerns.  The white-box assumption remains a key limitation, and no concrete defenses are proposed. Their discussion of related work is vague and incomplete. Overall, the concerns about novelty, practicality, and completeness still stand.

---

> ### Author Response · Authors · 2025-08-07
>
> Does the reviewer truly believe that the first *direct*, system-level attack on multimodal OS agents in the vision domain lacks novelty or practicality, especially at a time when such agents are gaining significant attention and are on the verge of public deployment?
>
> We find this position difficult to reconcile with the evidence. To our knowledge, **our work is the first to demonstrate that OS agents can be directly manipulated end-to-end on a system-level via MIPs**, under strict constraints, and with **robust transfer across *gray-box* conditions** involving unseen prompts, screen layouts, execution steps, sampling parameters, and OS agent components (see Tables 1–4 and Appendix Tables 8–9). Notably, our finding that MIPs do not transfer to entirely unseen VLM components in fact constitutes the currently best defense strategy, which we explicitly discuss in the paper.
>
> By identifying a new vulnerability of OS agents and rigorously characterising both its effectiveness and its limitations, we believe the work makes a foundational contribution to threat modeling, aligned with the structure and intent of prior impactful work in this area. This view is also supported by all other reviewers, who confirm that “this work is almost certain to inspire a series of follow-up research” (*LVfM*), that it “is of critical interest to the community” (*o8pU*), and that “the topic of security of OS agents is very crucial” (*8zsk*).
>
> Finally, we would gladly incorporate any related work the reviewer believes is missing. However, **without specific references, this critique cannot be addressed meaningfully**.
>
> We hope these clarifications help contextualise our contributions more clearly.

---

### Decision · Program_Chairs · 2025-09-17

**Decision:**

Accept (poster)

**Comment:**

The paper is slightly incremental in its methodology - applies well-known attack strategies and also studies a well-known attack vector (images). However, the paper applies all this in a new setting of OS agents and offers a few new analysis, mainly the robustness of their attacks to real-world settings. The paper overall seems comprehensive and well-executed. The reviewers raised various valuable concerns that the authors should systematically address in the camera ready version. Some of the concerns like white-box access were discounted in this meta review as they have been sufficiently addressed in the author response (evaluated in black-box settings as well).